# Extremely High-Quality Periodic Structures on ITO Film Efficiently Fabricated by Femtosecond Pulse Train Output from a Frequency-Doubled Fabry–Perot Cavity

**DOI:** 10.3390/nano13091510

**Published:** 2023-04-28

**Authors:** Qilin Jiang, Yuchan Zhang, Yufeng Xu, Shian Zhang, Donghai Feng, Tianqing Jia, Zhenrong Sun, Jianrong Qiu

**Affiliations:** 1State Key Laboratory of Precision Spectroscopy, School of Physics and Electronic Science, East China Normal University, Shanghai 200062, China; jiangqilin@tju.edu.cn (Q.J.);; 2Collaborative Innovation Center of Extreme Optics, Shanxi University, Taiyuan 030006, China; 3State Key Laboratory of Optical Instrumentation, Zhejiang University, Hangzhou 310027, China

**Keywords:** extremely high-quality LIPSS, frequency-doubled Fabry–Perot cavity, pulse train, ITO film, femtosecond laser

## Abstract

This study developed a novel frequency-doubled Fabry–Perot cavity method based on a femtosecond laser of 1030 nm, 190 fs, 1 mJ, and 1 kHz. The time interval (60–1000 ps) and attenuation ratio (0.5–0.9) between adjacent sub-pulses of the 515 nm pulse train were able to be easily adjusted, while the efficiency was up to 50% and remained unchanged. Extremely high-quality low-spatial-frequency LIPSS (LSFL) was efficiently fabricated on an indium tin oxide (ITO) film using a pulse train with a time interval of 150 ps and attenuation ratio of 0.9 focused with a cylindrical lens. Compared with the LSFL induced by the primary Gaussian pulse, the uniformity of the LSFL period was enhanced from 481 ± 41 nm to 435 ± 8 nm, the divergence of structural orientation angle was reduced from 15.6° to 3.7°, and the depth was enhanced from 74.21 ± 14.35 nm to 150.6 ± 8.63 nm. The average line edge roughness and line height roughness were only 7.34 nm and 2.06 nm, respectively. The depths and roughness values were close to or exceeded those of resist lines made by the interference lithography. Compared with the common Fabry–Perot cavity, the laser energy efficiency of the pulse trains and manufacturing efficiency were enhanced by factors of 19 and 25. A very colorful “lotus” pattern with a size of 30×28 mm^2^ was demonstrated, which was covered with high-quality LSFLs fabricated by a pulse train with optimized laser parameters. Pulse trains can efficiently enhance and prolong the excitation of surface plasmon polaritons, inhibit deposition particles, depress ablation residual heat and thermal shock waves, and eliminate high-spatial-frequency LIPSS formed on LSFL, therefore, producing extremely high-quality LSFL on ITO films.

## 1. Introduction

Laser-induced periodic surface structures (LIPSSs) have been of great interest and intensively studied over the past several decades. Several types of LIPSS have been prepared in various materials irradiated with femtosecond lasers, such as metals, semiconductors, and transparent materials [1,2,3,4]. LIPSS with a period λ_in_/2 < Λ < λ_in_, where λ_in_ is the wavelength of the incident laser, is called a low-spatial-frequency LIPSS (LSFL) [3]. The scattering model and surface plasmon polariton (SPP) model have been intensely studied and explained the formation of LSFL [1,4].

When Λ < λ_in_/2, the LIPSS is called a high-spatial-frequency LIPSS (HSFL) [3,5]. However, the formation mechanism of HSFL remains an interesting problem. Thin layers [6], local field enhancement [7], and high harmonics models [8] have been proposed to explain the formation of HSFL perpendicular to the laser polarization. HSFL parallel to the laser polarization can be induced alone or simultaneously with LSFL perpendicular to the laser polarization. Ionin et al. proposed that the HSFL parallel to the laser polarization was caused by spalling-cooling or eruption owing to the coherent cavitation instability of the molten surface layer [9].

Parallel HSFL appeared on the perpendicular LSFL and hindered the fabrication of high-quality LSFLs, which is a common phenomenon in semiconductors, metals, and transparent materials [6,10,11,12]. LIPSS is important for fabricating subwavelength structures on various materials and has important applications in structural color, optical storage, absorption enhancement, and optoelectronic property modulation [10,13,14,15,16,17], etc. LIPSSs are flexible and efficient low-cost methods. A fundamental goal of femtosecond LIPSS is to ensure that the quality of the lines is close to, or even up to, the standard of lines of resistance in advanced chip lithography, which should simultaneously meet the following requirements: large depth, fine line edge roughness, excellent direction, and uniformity. The quality of LIPSS significantly affects its functional characteristics [14,18]. Therefore, improving the quality of LIPSSs is a fundamental research topic in this field [19].

Indium–tin oxide (ITO) films with a wide bandgap of 3.3–4.3 eV provide high electrical conductivity (~10^−6^ Ω·m) and transmittance (>85%) in the visible and near-infrared (NIR) wavelengths [20,21]. Numerous carriers are available because Sn^4+^ can be doped by replacing In^3+^. Therefore, it is the most widely used material in various optoelectronic devices. Fabricating high-quality LSFL on the surface of ITO films can effectively modulate the optoelectronic properties and increase transmission in the NIR band [10,22,23]. 

This study developed a novel frequency-doubled Fabry–Perot cavity that efficiently generates femtosecond laser pulse trains. The time interval and attenuation ratio between adjacent sub-pulses can be easily adjusted. Pulse trains efficiently enhance SPP excitation, and inhibit the ablation of residual heat and deposition particles; therefore, they eliminate HSFL formed on LSFL and produce high-quality LSFL on ITO film. When the sample was irradiated by a primary Gaussian pulse, the HSFL parallel to the polarization appeared simultaneously with the LSFL perpendicular to the polarization, and the presence of HSFL significantly reduced the quality of LSFL. The average depth of LSFL induced by the primary Gaussian pulse was only 74.21 nm, whereas the depth of LSFL induced by the pulse train increased to 150.60 nm. Moreover, the line -height roughness of LSFL was depressed from 15.54 nm to 2.06 nm, and the divergence of structural orientation angle (DSOA) was reduced from 15.58° to 3.7°. The line edge roughness was only 7.34 nm on average. The experimental results demonstrate that large-area, high-quality LSFL can be efficiently fabricated using pulse trains with optimized laser parameters.

## 2. Experimental Setup and Method

### 2.1. Experimental Setup of Laser Direct Writing of LIPSS

Figure 1 shows the experimental system for the direct writing of LIPSS using femtosecond laser pulse trains based on a frequency-doubled Fabry–Perot cavity. A femtosecond laser (Pharos, Light Conversion, Vilnius, Lithuania) was used to generate laser pulses with a central wavelength of 1030 nm, pulse duration of 190 fs, repetition frequency of 1 kHz, and single-pulse energy of 1 mJ. A femtosecond laser pulse train with an adjustable time interval and attenuation ratio between adjacent sub-pulses was generated using a frequency-doubled Fabry–Perot cavity described in Figure 2. The focal length of the cylindrical lens used in the experiment was 50 mm and the focal spot was 15 μm (1/e^2^) wide. The samples were placed on an electronically controlled three-axis translational stage (XYZM118-150D, Lianyi Inc, Shanghai, China). The inset in Figure 1 shows the laser focus, polarization, and scanning direction. A charge-coupled device (VTSE3S-600, VIHENT, Shanghai, China) was used to monitor the laser processing in real time.

### 2.2. Method of Generating Pulse Trains in Frequency-Doubled Fabry–Perot Cavity

Pulse trains can be generated via several methods. Mastellone et al. reported two-dimensional LIPSSs on diamond surfaces irradiated with two temporally delayed and cross-polarized femtosecond laser pulses generated with a Michelson-like interferometer configuration [24]. Cong et al. reported a two-dimensional matrix of subwavelength dot structures directly generated on molybdenum surfaces by two-color femtosecond laser pulses [25]. Zhang et al. reported extremely regular LIPSSs fabricated via femtosecond pulse trains generated by a 4f pulse shaping system [14]. The Fabry–Perot cavity is a simple and stable method for generating pulse trains [26]. However, the very low efficiency of the laser power is one of the biggest disadvantages, which depends on the product of the transmittances of the two beam splitters. For example, if the expected sub-pulse energy attenuation ratio is 0.9, the reflectance of the two beam splitters should be ~95%, and the transmittance should be only ~5%. The laser utilization efficiency was calculated to be only 2.63%, where the utilization efficiency is the ratio of the output laser power from the Fabry–Perot cavity and the incident laser power. We developed a novel pulse train generation system to improve laser utilization efficiency, as shown in Figure 2a. The frequency-doubled Fabry–Perot cavity has a dichroic mirror (transmittance > 99% at a wavelength of 1030 nm and reflectance > 99% at a wavelength of 515 nm at an incidence angle of 0°), BBO (effective at a wavelength of 1030 nm with a maximum efficiency of 50%), and a beam splitter (effective at a wavelength of 515 nm).

A beam analyzer (BC106N-VIS/M, Thorlabs, Newton, NJ, USA) was installed at the cylindrical lens’ focal point to monitor the sub-pulses’ spatial overlap. The laser utilization efficiency reached a maximum of 50%, determined only by the frequency-doubling efficiency and not the expected attenuation ratio of the sub-pulses. For the expected sub-pulse energy attenuation ratio of 0.9, the laser utilization efficiency increased by 19 times, which is conducive to efficiently manufacturing high-quality LSFL in large areas.

The beam splitter was mounted on a two-dimensional electronically controlled mirror holder (8807, MKS Instruments Inc., Andover, MA, USA), which was used to adjust the spatial overlap of the pulse train. The mirror holder was mounted on a one-dimensional translation stage (XM118-150D, Lianyi Inc, Shanghai, China), which was used to control the pulse train’s time interval (in the range of 60–1000 ps) by changing the distance between the dichroic mirror and the beam splitter. The energy attenuation ratio of the pulse train was controlled by selecting beam splitters with different splitting ratios. Figure 2b shows the theoretically calculated energy distribution of pulse train output from the frequency-doubled Fabry–Perot cavity. The sub-pulse energy decreased in an attenuation ratio of 0.9/0.7/0.5 when the transmittance (T) verse reflectance (R) ratio of the beam splitter was 1:9/3:7/5:5, respectively. To more clearly compare the decay trend of pulse trains with different attenuation ratios, the energy of the first sub-pulse for every ratio was assumed to be the same in calculation, and the energy of sub-pulse was normalized by the energy of the first sub-pulse in every ratio as well.

It is important to consider how many sub-pulses could affect the material before the energy is too low to cause any alteration of the surface, that is, how many effective sub-pulses would be required in order to explore how the pulse train induces regular LIPSS. Firstly, the number of effective sub-pulses is determined by the sub-pulse energy itself. For example, if the energy of the first sub-pulse is the same, a pulse train with an attenuation ratio of 0.9 has more effective sub-pulses than those of 0.7 and 0.5 for the lower attenuation ratio. Secondly, the number of effective sub-pulses is influenced by the time interval between sub-pulses. As the time interval between sub-pulses increases, material ablation, thermal radiation, and heat conduction weaken the continuous excitation effect of the subsequence pulses, resulting in fewer effective sub-pulses.

We suggest preparing ultrafast luminescence experiments to explore this critical problem. When irradiating the sample with a pulse train, a streak camera can be used to detect plasma emission. Each sub-peak of plasma emission can be clearly observed and effective sub-pulses are demonstrated accurately.

### 2.3. Sample Preparation and Measurement

The sample used in the experiments was a commercial ITO film (ITO-400, Xiangcheng Technology Inc., Dongguan, China) with a thickness of 400 nm, uniformly coated on a quartz glass substrate with a surface roughness of 1 nm by magnetic sputtering. The sample was ultrasonically cleaned using different solvents before laser irradiation. It was washed with a 10% acetone solution for 15 min to remove surface oil stains and then with a 10% ethanol solution for 20 min to remove residual acetone from the surface. It was then washed with a large amount of deionized water for 30 min to remove the surface ethanol.

The surface morphology of ITO film after the laser irradiation was examined using a scanning electron microscope (SEM; S-4800, HITACHI, Tokyo, Japan), and the depth was measured using an atomic force microscope (AFM; NanoWizard II, Bruker, Billerica, MA, USA). The structural color patterns were photographed using a CCD camera (D5500, Nikon, Tokyo, Japan).

## 3. Results and Discussion

### 3.1. LIPSS Induced on ITO Film by Primary Gaussian Pulse

The primary Gaussian pulse was a femtosecond pulse at a central wavelength of 515 nm after frequency doubling of the 1030 nm laser but without passing through the Fabry–Perot cavity. By varying the laser fluence and scanning velocity, Figure 3 shows that three main types of surface morphologies dominated by LIPSS were formed on the surface of ITO film: (a) HSFL (para) parallel to the polarization direction, (b) continuous LSFL (perp) perpendicular to the polarization direction covered with HSFL (para), and (c) severely damaged LSFL (perp) covered with HSFL (para). Figure 3b,d,f show Fourier transform images of the corresponding surface morphologies of Figure 3a,c,e. In Figure 3b, only a distinct component along the k_y_ direction existed in the Fourier transform image, corresponding to the HSFL (para) with a period of 173 ± 13 nm. In Figure 3d, the components exist along both the k_x_ and k_y_ directions, indicating that LSFL (perp) coexists with HSFL (para). In Figure 3f, LSFL (perp) and HSFL (para) still coexist on severely damaged surfaces.

Figure 4 shows the three morphologies of ITO film obtained by changing the laser fluence and scanning velocity under primary Gaussian pulse irradiation. When the scanning velocity was ≤0.6 mm/s, HSFL (para) and LSFL (perp) simultaneously appeared on the surface of the ITO film. They were gradually damaged with an increase in laser fluence. When the scanning velocity was in the range of 0.7–0.8 mm/s, only HSFL (para) was induced at low laser fluence, and the surface evolved to the morphology of coexisting LSFL (perp) and HSFL (para) after gradually increasing the laser fluence. When the scanning velocity increased above 0.8 mm/s, HSFL (para) appeared at low laser fluence, while damaged HSFL (para) coexisted with LSFL (perp) at higher laser fluence. In a word, because of the serious mutual disturbance between HSFL (para) and LSFL (perp), no regular LSFL or HSFL was formed on ITO film, regardless of the laser scanning velocity and laser fluence.

### 3.2. Extremely High-Quality LSFL Induced on ITO Film by Pulse Train

When the ITO film was irradiated by a pulse train with an attenuation ratio of 0.9 and a time interval of 150 ps between adjacent sub-pulses, it was astonishing that extremely high-quality LSFL (perp) was formed on the ITO film. The HSFL (para) completely disappeared, as shown in Figure 5a, where the laser fluence was 0.37 J/cm^2^, and the scanning velocity was 0.5 mm/s. Laser fluence was measured in front of the sample, which was the same as the primary Gaussian pulse. Due to the presence of the frequency-doubled Fabry–Perot cavity, there were many sub-pulses in the pulse train, and 0.37 J/cm^2^ was the total laser fluence of all sub-pulses. In the corresponding Fourier transform image in Figure 5b, only the components along the k_x_ direction and no components along the k_y_ direction were observed, indicating that HSFL (para) was not formed. The Fourier transform peak occurred at 2.300 ± 0.040 μm^−1^, corresponding to a period of 435 ± 8 nm. The divergence of structure orientation angle (DSOA) is the half-width value of the angular distribution at half of the highest intensity value, which is used to analyze the bending and directivity of LSFLs [14]. As shown in Figure 5a, the DSOA of LSFL was very small, at only 3.7°. Such small fluctuations in the period and DSOA further indicate that the LSFL was extremely uniform and straight.

The surface roughness and depth of LIPSS were further measured by atomic force microscopy (AFM). Figure 6a,d show AFM images of a 5 × 3 μm^2^ area of LIPSS induced by the pulse train with a time interval of 150 ps and attenuation ratio of 0.9, and the primary Gaussian pulse, respectively. The average depth of the LSFL induced by the pulse train was 150.60 nm, and the standard deviation of the depth was only 8.63 nm. The LSFL induced by the pulse train was very deep and uniform. In contrast, the average depth of LSFL (perp) induced by the primary Gaussian pulse was only 74.21 nm, and the standard deviation of the depth was 14.4 nm. The average depth of LSFL (perp) induced by the primary Gaussian pulse was less than half that of the pulse train, and the standard deviation was 1.66 times that of the pulse train. The surface roughness of LSFL (perp) was obtained by calculating its surface profile of LSFL in Figure 6a,d using the least-squares method, as shown in Figure 6c,f. The roughness of LSFL induced by the pulse train was only 2.06 nm, indicating the formation of smooth gratings on the ITO film. However, the roughness of LSFL (perp) induced by the primary Gaussian pulse reached 15.54 nm, 7.54 times that of the pulse train. The difference in roughness was mainly caused by HSFL (para) when irradiated by the primary Gaussian pulse.

The line edge roughness (LER) was calculated using high-magnification SEM images, which were used to characterize the flatness of the structural boundaries in the field of laser lithography. Here, the LERs of the LSFLs induced by the pulse train, with an attenuation of 0.9 and time interval of 150 ps, and a primary Gaussian pulse were analyzed. Ten clear edges of the LSFLs were obtained via image processing, as shown in Figure 7c,d. Table 1 shows that the LER of LSFLs induced by the pulse train with attenuation of 0.9 and time interval of 150 ps was 5.21 nm at the minimum and 9.12 nm at the maximum. The average value of LER was only 7.34 nm, close to the value of lines of resist produced by extreme ultraviolet lithography [27,28].

However, the surface morphology shown in Figure 3c did not define clear edges. For the damaged HSFL (para) and LSFL (perp) induced by the primary Gaussian pulse, Figure 7b,d show that the LER of LSFLs was 18.93 nm at the minimum and 47.57 nm at the maximum. The average LER was 33.05 nm, which was 4.5 times larger than that of the pulse train.

In summary, the fluctuations in the period, DSOA, and LER of LSFL induced by the pulse train were minimal, and the groove was very smooth and deep with a small standard deviation. Extremely high-quality LSFL was fabricated via pulse train with a time interval of 150 ps and attenuation ratio of 0.9.

### 3.3. Discussion: The Formation of Extremely High-Quality LSFL Induced by Pulse Train

The formation processes of LIPSSs induced by pulse trains and primary Gaussian pulses of ITO films are significantly different. When irradiated by the primary Gaussian pulse, the carriers are intensely excited and form a plasma layer on the ITO film’s surface, which supports SPP’s excitation and the formation of LSFL (perp). The duration of LSFL (perp) generation typically ranges from tens of picoseconds to hundreds of picoseconds [29]. The phase explosion, vaporization, and spallation of a part of the melt layer of materials always co-occur with the formation of LSFL (perp), but these processes last much longer, ranging from picoseconds to nanoseconds [30,31]. The thermal shock wave caused by spallative ablation not only partly erases the LSFL (prep), but also leads to coherent cavitation of the surface melting layer and the generation of HSFL (para) [9]. The HSFL (para) generation causes serious damage to the depth and regularity of LSFL (perp). Finally, the morphology of coexisting HSFL (para) and LSFL (perp), or even HSFL (para), form on the surface of the ITO film.

When a femtosecond laser pulse train with an appropriate time interval irradiates the ITO film, extremely high-quality LSFL (perp) is induced, and no HSFL (para) exists for the following reasons. Firstly, when the subsequent sub-pulse arrives, the heat transfer among lattices caused by the preceding sub-pulse is only in the beginning. The hot surface layer reaches the boiling point and evaporates directly after multiple heating by the sub-pulse, removing a large amount of residual heat, called the ablation cooling effect [32,33,34,35]. The ablation cooling effect reduces the thickness of the surface melting layer and the pressure inside the material. In addition, compared with the primary Gaussian pulse, the intensity of each sub-pulse is much lower; therefore, the carriers are moderately excited, and the thermal effect is much smaller. The smaller thermal shock wave and thinner surface melting layer significantly suppress or eliminate the HSFL (para) caused by coherent cavitation. Secondly, the ejected particles in the plume generated by the previous sub-pulse further absorb the subsequent sub-pulse laser energy, crack into smaller debris, and evaporate into gas. Fewer ejecta depositions are conducive to forming regular and smooth LSFL (perp) [36,37]. Thirdly, the SPP excited by the pulse train lasts much longer. For example, a pulse train with an attenuation ratio of 0.9 and a time interval of 150 ps lasts for more than 2 ns. SPP is repeatedly excited and significantly enhanced by a pulse train. Therefore, the formation of LSFL (perp) lasts for a long time, even throughout the entire process of ablation and melting [38], which significantly enhances the LSFL depth, improves the regularity and uniformity, and eliminates the HSFL (para).

### 3.4. Large-Area High-Quality LSFL Efficiently Fabricated on ITO Films via Pulse Train

High-quality LSFL over large areas is important for the application of ITO in electrodes. In the following sections, we study how to efficiently fabricate high-quality LSFL on ITO films using pulse trains with a central wavelength of 515 nm.

#### 3.4.1. The Attenuation Ratio of Adjacent Sub-Pulses

As shown in Figure 8, the fixed laser scanning velocity was 0.5 mm/s, and the time interval was 150 ps. LSFLs with different surface morphologies were fabricated using pulse trains with attenuation ratios of 0.5, 0.7, and 0.9, respectively. In Figure 8a,d,g, under the irradiation of a pulse train with an attenuation ratio of 0.5 and laser fluence of 0.27 J/cm^2^, LSFL (perp) with a period of 460 ± 21 nm was formed on the ITO film. Compared with the nanostructures induced by the primary Gaussian pulse, as shown in Figure 3c, LSFL (perp) became regular, and the Fourier transform peak of HSFL (para) was very weak and could not be observed. The HSFL (para) was suppressed by the pulse train and became irregular; severe cracks covered the LSFL (perp), causing the LSFL (perp) to be very coarse. The LSFL (perp) was improved by the pulse train with an attenuation ratio of 0.7, as shown in Figure 8b,e,h. When the attenuation ratio was increased to 0.9 with a laser fluence of 0.33 J/cm^2^, LSFL (perp) was very regular, with a smooth profile and clear straight edges, as shown in Figure 8c,f. The Fourier transform image showed that the period of LSFL was 435 ± 8 nm, with minimal fluctuation. Figure 2 shows two, three, and eight efficient sub-pulses (half the maximum intensity) for pulse trains with attenuation ratios of 0.5, 0.7, and 0.9, respectively. These results showed that more effective sub-pulses are beneficial for inhibiting the HSFL (para) and thermal effects, and for prolonging and enhancing the excitation of SPP. All these factors work together, leading to the formation of high-quality LSFL (perp).

In the experiments, laser fluences fabricating better LIPSS continuously increased as the attenuation ratio increased from 0.5 to 0.9, because the pulse train with an attenuation ratio of 0.9 had the lowest sub-pulse energy when total laser fluences stayed the same compared to 0.5 and 0.7. The suitable laser fluences were 0.27 J/cm^2^, 0.29 J/cm^2^ and 0.33 J/cm^2^ when the attenuation ratio was 0.5, 0.7 and 0.9, respectively. Demonstrating LIPSS induced by suitable laser fluence, rather than the same fluence, allows for a clearer comparison of the impact of attenuation ratio on the quality of LIPSS.

#### 3.4.2. Time Interval between Adjacent Sub-Pulses

In addition to the attenuation ratio of the pulse train, the LSFL formation process is intensely affected by the time interval between adjacent sub-pulses. Figure 9 shows the surface morphologies of the LSFLs with relatively better quality induced by pulse trains with different time intervals, where the attenuation ratio was 0.9 and the scanning velocity was 0.5 mm/s. Due to the ultrafast process of laser fabricating LSFL, the proper laser fluence of fabricating LSFL with relative better quality was different when the time interval significantly varied [39]. The quality of LSFL varied greatly with the time intervals. When the time interval was less than 200 ps, the LSFLs exhibited smooth profiles and good regularity. At a time interval of 200 ps, the edges of the LSFLs were slightly distorted. When the pulse interval was increased to 300–400 ps, the LSFL was clear; however, the many cracks covering the ridge made the LSFL very coarse. When the pulse interval exceeded 600 ps, the LSFL became blurred and coarse. If the time interval between adjacent sub-pulses is too long, the ablation cooling effect is significantly weakened, and the particles deposited on ITO film significantly increases. Therefore, the quality of LSFL became very poor.

Figure 10 shows that the DSOA quantitatively characterizes the regularity of LSFLs. The DSOA of LSFL induced by the primary Gaussian pulse was 15.6°. As the time interval increased from 60 ps to 150 ps, the DSOA reached a minimum value of 3.7°. Subsequently, the DSOA continued to increase, reaching 10.9° at 600 ps. 

These experimental results show that regular LSFLs (perp) with smooth profiles and smaller orientation angles can be fabricated using pulse trains with a time interval of ~150 ps.

#### 3.4.3. Laser Fluence and Scanning Velocity

The attenuation ratio and time interval between adjacent sub-pulses were kept at 0.9 and 150 ps, respectively. The following section discusses the effects of the laser fluence and scanning velocity on the quality of LSFL on ITO film. 

Figure 11a shows that a discontinuous LSFL (perp) was first induced at a low laser fluence of 0.30 J/cm^2^. At an appropriate laser fluence of 0.33 J/cm^2^, as shown in Figure 11b, a high-quality LSFL (perp) was formed on ITO film. When the laser fluence was further increased to 0.41 J/cm^2^, LSFL (perp) was severely damaged and became very coarse and curved, as shown in Figure 11c. The scanning velocity was kept at 0.5 mm/s.

Figure 11d shows the ranges of laser fluence and scanning velocity corresponding to the three main morphologies of LSFL (perp). In the range of 0.1–0.7 mm/s, discontinuous, regular, and damaged LSFL (perp) was induced sequentially with the increase of laser fluence. However, when the scanning velocity was above 0.7 mm/s, only discontinuous LSFL (perp) and damaged LSFL (perp) were formed on the ITO film, indicating that proper cumulative laser pulse irradiation was essential to fabricate high-quality LSFL. The results demonstrate that high-quality LSFLs (perp) could be fabricated by laser direct writing with a pulse train, where the laser fluences are in the green region, and the scanning velocity in the range of 0.2–0.6 mm/s.

The efficient fabrication of large-area high-quality LSFL is essential for its applications. In our experimental system, the laser pulse energy was 1.0 mJ, 190 fs, 1030 nm, and 1 kHz, which can generate pulse trains with a power of 0.5 W at a central wavelength of 515 nm. The laser was irradiated perpendicularly onto the ITO film using a cylindrical lens with a focal length of 50 mm. Using a cylindrical lens can improve the efficiency of laser processing and the regularity of the LSFL [18]. When the laser fluence was 0.37 J/cm^2^ and the scanning velocity was 0.5 mm/s, an ablation band of 0.5 mm wide covered with high-quality LSFL (perp) was fabricated on the ITO film by single laser direct writing. If a conventional Fabry–Perot cavity with only two beam splitters is used to generate a pulse train, the laser utilization efficiency is only 2.63% if a pulse attenuation ratio of 0.9 is achieved. Focused by a circular lens with a focal length of 50 mm, only 20 μm width of LSFL can be processed. Therefore, with the pulse-shaping method proposed here, the efficiency of processing high-quality LSFLs is increased by 25 times. In addition, more regular LSFLs are produced than those processed with a circular lens.

The “lotus” pattern with an area of 30 × 28 mm^2^ was efficiently fabricated in 60 min by laser direct writing. Figure 12 shows that extremely high-quality LSFL covered the entire lotus pattern. In a dark background and under illumination by an LED light source, the lotus pattern showed excellent purity and bright color [40,41,42].

## 4. Conclusions and Outlook

In summary, we proposed a novel frequency-doubled Fabry–Perot cavity method to generate femtosecond pulse trains with a central wavelength of 515 nm and an adjustable time interval and attenuation ratio between adjacent sub-pulses. The efficiency was up to 50% and remained unchanged when the attenuation ratio changed in the range of 0.5–0.9. Extremely high-quality LSFL was efficiently fabricated on ITO film using a pulse train with an interval of 150 ps and an attenuation ratio of 0.9, focused with a cylindrical lens. The DSOA of LSFL was reduced to 3.7°; the depth was enhanced to 150.6 ± 8.63 nm, and the average line edge roughness and line height roughness were only 7.34 nm and 2.06 nm, respectively. All these factors were close to or exceeded the lines of resistance made by interference lithography. Compared with the Gaussian pulse, pulse trains can efficiently enhance and prolong the excitation of SPP, reduce the residual heat, weaken the thermal shock wave, and eliminate HSFL, thereby fabricating extremely high-quality LSFL on ITO film.

Compared with the common Fabry–Perot cavity, the laser energy efficiency of the pulse trains was enhanced by a factor of 19, and the manufacturing efficiency of LSFL by a factor of 25, respectively. In addition, a very colorful “lotus” pattern with a size of 30 × 28 mm^2^ was demonstrated, which was covered with high-quality LSFLs fabricated by a pulse train with optimized laser parameters. LIPSS is a fast nanofabrication and maskless technique that has drawn increasing attention for its applications in various fields. The experimental results demonstrate that this method can effectively fabricate high-quality LSFLs using pulse trains. By conducting more experiments on different types of materials, this method will lead to the further development of LIPSS.

## Figures and Tables

**Figure 1 nanomaterials-13-01510-f001:**
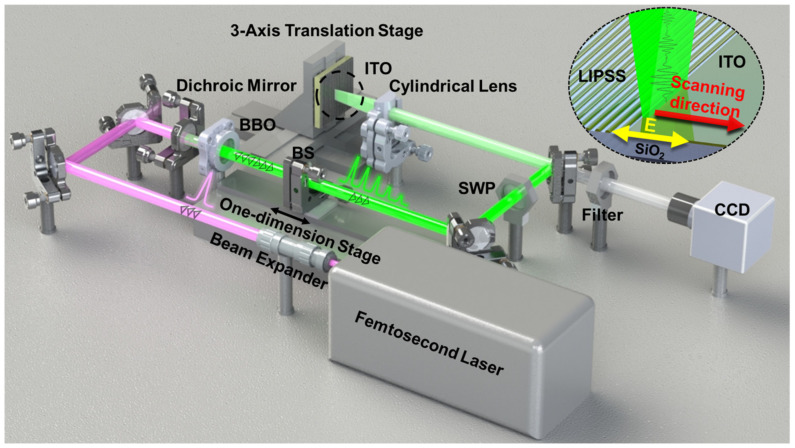
Experimental system for direct writing of LIPSS by femtosecond laser pulse trains on ITO film based on a frequency-doubled Fabry–Perot cavity. BS: beam splitter. SWP: short wave pass. CCD: charge-coupled device. BBO: barium metaborate crystal. The dashed inset shows the scanning direction and polarization direction of the laser.

**Figure 2 nanomaterials-13-01510-f002:**
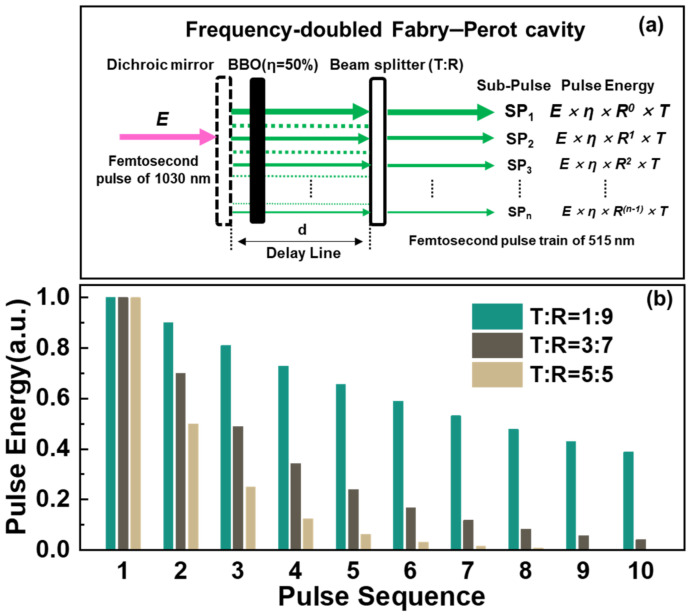
(**a**) Schematic of frequency-doubled Fabry–Perot cavity. *E* is the energy of the incident laser pulse with a central wavelength of 1030 nm. *η* is the efficiency of BBO. *T* and *R* represent the transmittance and reflectance of the beam splitter. (**b**) Sub-pulse energy calculated with different attenuation ratios of beam splitters with transmittance (T) verse reflectance (R) ratios of 1:9/3:7/5:5, respectively. Energy is normalized by the energy of the first sub-pulse in every ratio.

**Figure 3 nanomaterials-13-01510-f003:**
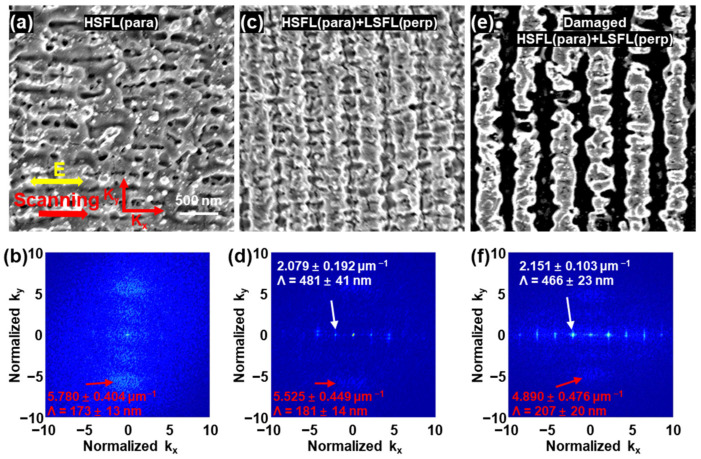
(**a**,**c**,**e**) SEM images of the three main types of surface morphologies on the surface of ITO films irradiated by primary Gaussian pulse; the polarization and scanning direction in (**a**) are suitable for (**c**,**e**). The scale bar is 500 nm. (**b**,**d**,**f**) Fourier transform images corresponding to the three morphologies in (**a**,**c**,**e**), respectively.

**Figure 4 nanomaterials-13-01510-f004:**
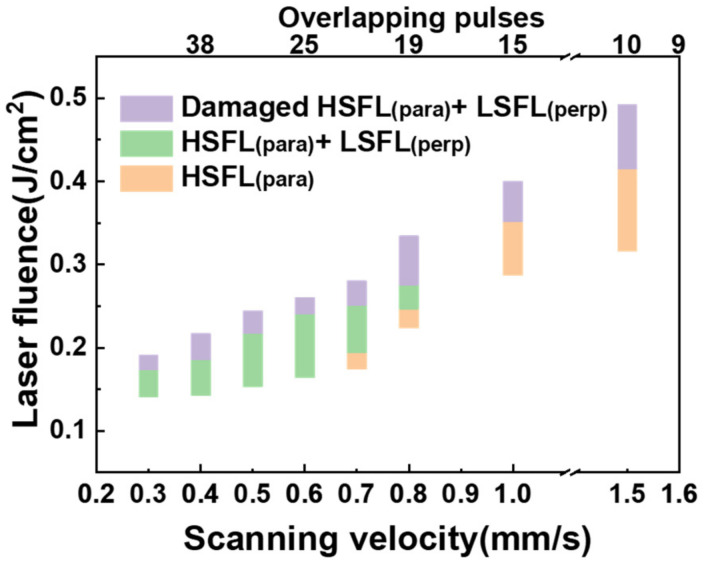
Laser fluence and scanning velocity corresponding to the three main morphologies of ITO surfaces after irradiation of primary Gaussian pulse.

**Figure 5 nanomaterials-13-01510-f005:**
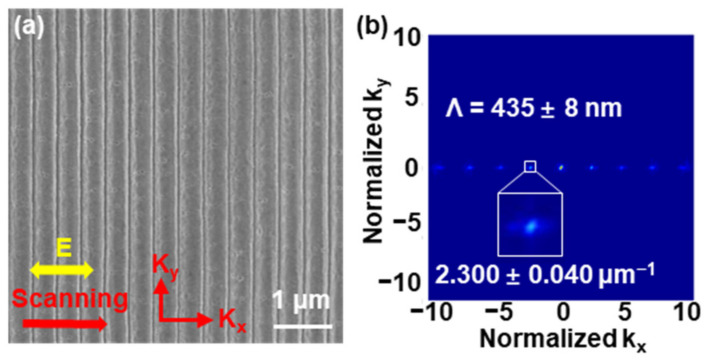
(**a**) SEM image of the surface morphologies on the surface of ITO films irradiated by pulse train, and (**b**) the Fourier transform image.

**Figure 6 nanomaterials-13-01510-f006:**
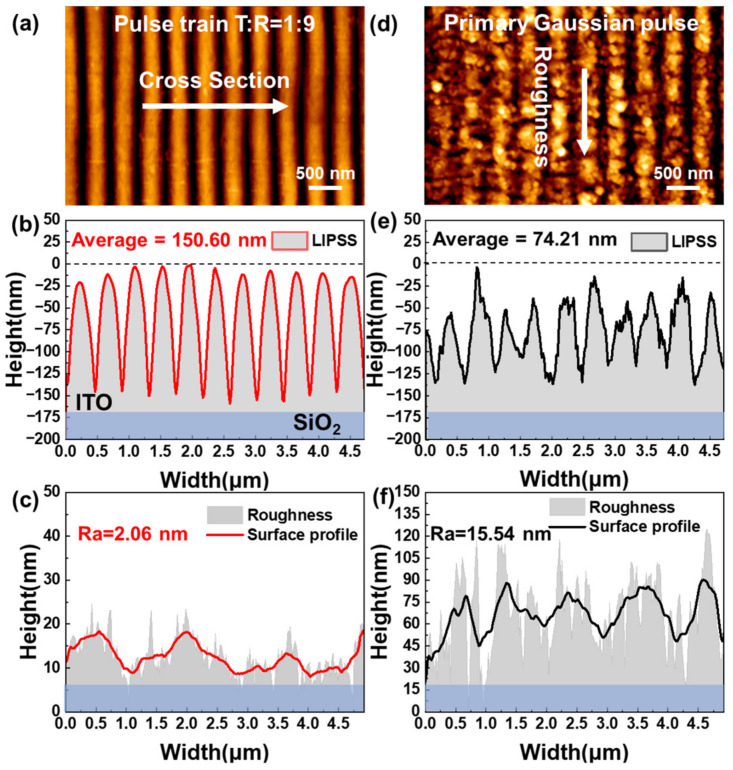
Depth and roughness of LIPSS induced by pulse train and primary Gaussian pulse. (**a**–**c**) High-magnification AFM image of the LSFL in Figure 5a and (**d**–**f**) AFM image of the LSFL (perp) and HSFL (para) in Figure 3c.

**Figure 7 nanomaterials-13-01510-f007:**
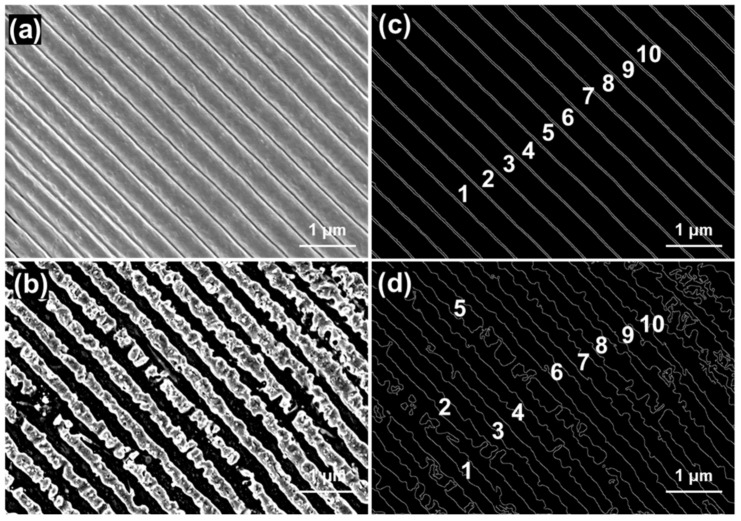
SEM images of LSFL induced by the (**a**) pulse train and (**b**) primary Gaussian pulse. (**c**,**d**) Edges of grey level calculated via the image processing methods of (**a**,**b**), respectively.

**Figure 8 nanomaterials-13-01510-f008:**
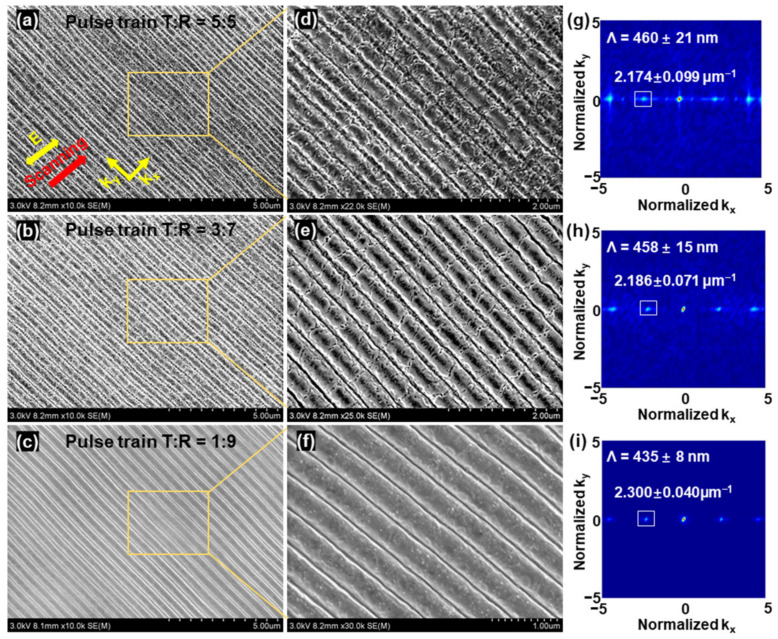
LSFL induced by pulsed trains on the surface of ITO films with different attenuation ratios. (**a**,**d**,**g**) SEM images and Fourier transform image of LSFL induced by pulse trains with an attenuation ratio of 0.9, (**b**,**e**,**h**) by pulse trains with attenuation ratio of 0.7, (**c**,**f**,**i**) by pulse trains with attenuation ratio of 0.5.

**Figure 9 nanomaterials-13-01510-f009:**
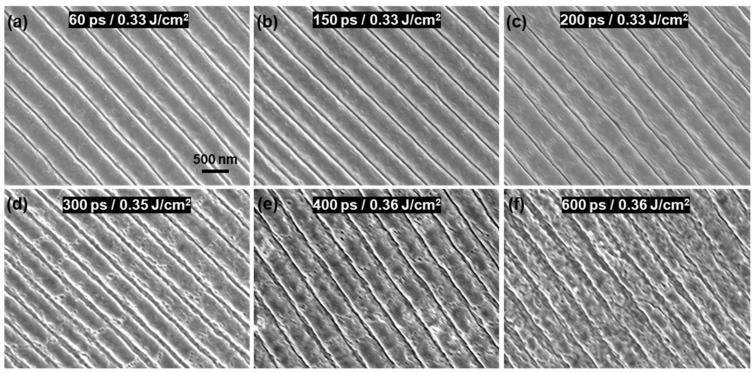
Surface morphologies of LSFLs induced via pulse train with time intervals of (**a**) 60 ps, (**b**) 150 ps, (**c**) 200 ps, (**d**) 300 ps, (**e**) 400 ps and (**f**) 600 ps, respectively. Laser fluences are 0.33 J/cm^2^ in (**a**–**c**), 0.35 J/cm^2^ in (**d**) and 0.36 J/cm^2^ in (**e**,**f**), respectively. The scale bar is 500 nm.

**Figure 10 nanomaterials-13-01510-f010:**
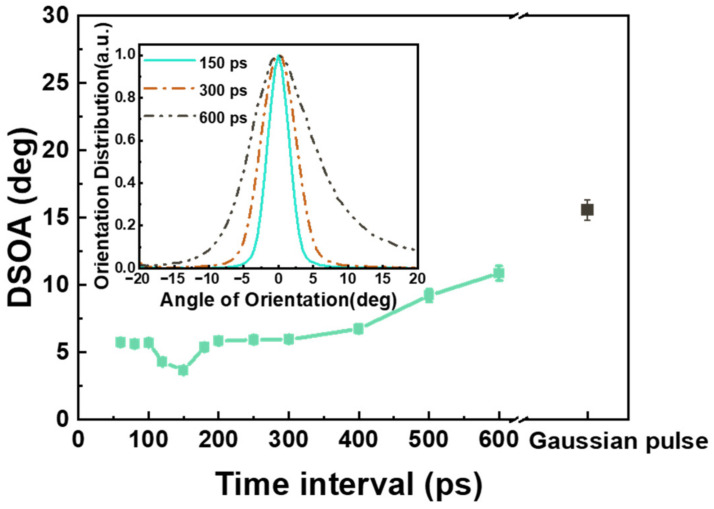
The variation curves of the DSOA of LSFLs induced by pulse trains with different time intervals. The inset shows the distribution of orientation angles of LSFLs induced by pulse trains with time intervals of 150 ps, 300 ps, and 600 ps, respectively.

**Figure 11 nanomaterials-13-01510-f011:**
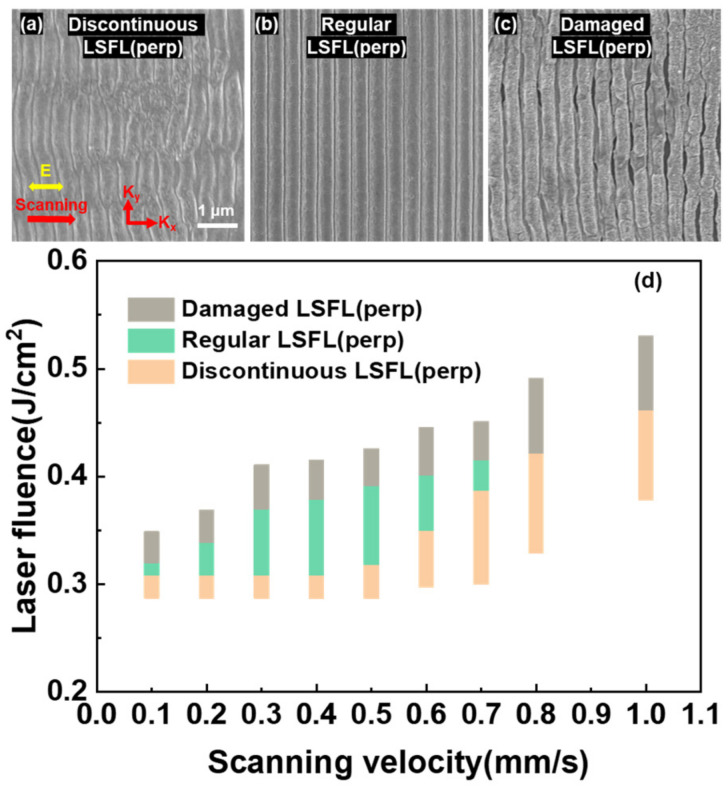
(**a**–**c**) The three main morphologies of the discontinuous, high-quality, and damaged LSFL (perp), respectively. (**d**) The range of laser fluence and scanning velocity corresponding to the three main morphologies of LSFL (perp).

**Figure 12 nanomaterials-13-01510-f012:**
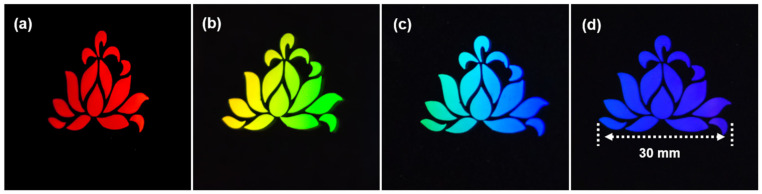
(**a**–**d**) The vivid colorful patterns under the illumination of the LED light source. The laser fluence is 0.37 J/cm^2^ at a scanning velocity of 0.5 mm/s.

**Table 1 nanomaterials-13-01510-t001:** LERs of LSFL edges.

Line	1	2	3	4	5	6	7	8	9	10	Avg
Pulse train	5.40	5.21	7.00	7.65	9.04	8.65	7.48	6.82	7.20	9.12	7.34
Gaussian	43.66	31.22	47.57	32.29	27.61	34.78	28.59	23.66	18.93	42.23	33.05

Pulse train: LERs of LSFL induced by pulse train as shown in Figure 3a,c. Gaussian: LERs of LSFL induced by primary Gaussian pulse as shown in Figure 3b,d. Unit of LER: nm.

## Data Availability

Not applicable.

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
