# Peer review of "Extremely High-Quality Periodic Structures on ITO Film Efficiently Fabricated by Femtosecond Pulse Train Output from a Frequency-Doubled Fabry–Perot Cavity"

_nanomaterials, 2023, doi:10.3390/nano13091510_

Round 1

Reviewer 1 Report

This paper reports on the morphological characterization of Laser Induced Periodic Surface Structures (LIPSS) on thin Indium-tin oxide (ITO) films. The LIPSS are fabricated using a novel technique involving a frequency-doubled Fabry-Perot cavity that allows the formation of high-regular structures. The regularity was assessed using the calculation of the Fourier spectra of SEM images using two-dimensional (2D) Fourier transformation.

General concerns

The results obtained by the authors are notable and presents elements of novelty that concern the fabrication of high-regular LIPSS on large areas and for this reason it helps the further development of this surface altering technique. In addition, the topic of LIPSSs is still of great interest to the scientific community, and the problem of obtaining uniform structures is one of the main concerns that is being currently debated.

However, the paper presents some flaws on the methodological discussion that I will highlight in the next section. Both the description of the experiment and the presentation of the data are at time confusing and presents inaccuracies. For these reasons I suggest publication only after addressing the major issues that were spotted during the peer review process.

Thus, major revision is recommended before publication in Nanomaterials.

Major concerns

1.  The most glaring issue I have found with the paper is related to the lack of a scientific sound description of the sub-pulse train produced by the Fabry-Perot cavity. It is, in fact, quite difficult to extract the energies of the of sub-pulses composing the train from figure 2(b). As I understood the graph (and I may have misinterpreted the meaning of it) it shows the rate of the attenuation for every sequence. If that is the case, the label Pulse Energy on the y-axis is misleading, otherwise the sum of the pulses energy (for ratio 1:9 and 3:7) would exceed 1 (the starting energy).

2.  Additionally, it is not entirely clear how many sub-pulses will affect the material before the energy will be too low to cause any alteration of the surface, or to continuously “feed” the initial excitation.

3.  From section 3.2 onwards, when the alterations from the sub-pulses train are analysed, it is reported a laser fluency parameter that is not explicated (row 189). Is that the same value as the primary Gaussian Pulse? So, the fluence before the cavity? Or is it the value of the highest energetic sub-pulse?

4.  Since there were used three different attenuation ratios, it would be interesting to know if the authors have calculated how many sub-pulses will affect the creation of LIPSS. As an example, I presume the attenuation ratio 5:5 would be comprised by more over-threshold sub-pulses compared to the other 2 situations.

5.  It is not explained why the average depth of LSFL is higher when the Fabry-Perot cavity is used.

6.  Another important point relates section 3.4.1. Why does the laser fluence change while the effect of attenuation ratios are analysed? It goes from 0.267 J/cm2 for 5:5 and 3:7 to 0.329 J/cm2 for 1:9. Why?

7.  Same thing happens in section 3.4.2. The fluence energy increases as the delay does. Why? And is it possible to disentangle the effect of the two parameters?

Minor concerns

1.  The abstract contains a discussion about LIPSS that are more suited to be in the introduction section (row 13-16). Furthermore, the laser parameters are not labelled but just quantities are reported.

2.  In the introduction, the discussion related to the HSFL is rather confusing. As an example, what is the thin-layer theory reported on row 46? Also, vertical HSFL are discussed, but what does vertical mean?

3.  Since the authors in the text highlight the possibility to obtain very narrow structures (lower than 10 nm, and comparable to extreme ultraviolet lithography), it would be fitting to also report on other deep-subwavelength LIPSS obtained using different techniques to showcase the potential and the novelty of the Fabry-Perot cavity. Some examples of deep-subwavelength structures:

a.  https://doi.org/10.1021/acs.nanolett.1c01310

b.  https://doi.org/10.1364/OE.23.005357

4.  In the caption of Figure 1, the BBO labelling is missing.

5.  It is not entirely clear how the laser utilization efficiency is calculated (row 105 and 106).

6.  The polarization and scanning direction of figure 3 are not entirely visible.

7.  Have the authors also considered analyzing the width of the distribution of the spatial frequencies of 2D-FFT images? In reference n.19 the authors present, in addition to DLOA, the “dispersion” of the spatial frequencies as a characterizing tool for LIPSS’ regularity. Do the fabricated Mo samples present differences in spatial frequencies width?

Author Response

Comments to the Author

This paper reports on the morphological characterization of Laser Induced Periodic Surface Structures (LIPSS) on thin Indium-tin oxide (ITO) films. The LIPSS are fabricated using a novel technique involving a frequency-doubled Fabry-Perot cavity that allows the formation of high-regular structures. The regularity was assessed using the calculation of the Fourier spectra of SEM images using two-dimensional (2D) Fourier transformation.

Dear referee, thanks very much to you to review our manuscript, and give us a chance to revise it. Your comments are very important for us to improve this manuscript. According to your suggestions, we made revisions in the resubmitted manuscript, which is shown in red words.

Best wishes,

All of the authors

The following major revision comments are suggested,

1) The most glaring issue I have found with the paper is related to the lack of a scientific sound description of the sub-pulse train produced by the Fabry-Perot cavity. It is, in fact, quite difficult to extract the energies of the of sub-pulses composing the train from figure 2(b). As I understood the graph (and I may have misinterpreted the meaning of it) it shows the rate of the attenuation for every sequence. If that is the case, the label Pulse Energy on the y-axis is misleading, otherwise the sum of the pulses energy (for ratio 1:9 and 3:7) would exceed 1 (the starting energy).

Thanks very much for your suggestions. Each energy of sub-pulse is calculated but not measured in figure 2(b), energy is normalized by the energy of the first sup-pulse in each ratio. We have made changes to figure 2(a) in order to show clearly the calculation process of each sub-pulse energy.

Figure 2. (a) Schematic of frequency-doubled Fabry–Perot cavity. E is the energy of the incident laser pulse with a central wavelength of 1030 nm. η is the efficiency of BBO.T and R represent the transmittance and reflectance of the beam splitter. (b) Sup-pulse energy calculated with different attenuation ratios of beam splitters with transmittance (T) verse reflectance (R) ratios of 1:9/3:7/5:5, respectively. Energy is normalized by the energy of the first sup-pulse in each ratio.

2) Additionally, it is not entirely clear how many sub-pulses will affect the material before the energy will be too low to cause any alteration of the surface, or to continuously “feed” the initial excitation.

Thanks very much for your suggestions. Your suggestion is very important and useful to explore how the pulse train induces regular LIPSS. First, the number of effective sub-pulses is determined by the sub-pulse energy itself. For example, if the energy of the first sub-pulse is the same, pulse train with an attenuation ratio of 0.9 has more effective sub-pulses than those of 0.7 and 0.5 for the lower attenuation ratio. Second, the number of effective sub-pulses is influenced by the time interval between sub-pulses. As the time interval between sub-pulses increases, material ablation, thermal radiation, and heat conduction weaken the continuous excitation effect of the subsequence pulses, resulting in fewer effective sub-pulses.

We are preparing experiments to explore this key problem. We plan to use a streak camera to probe the plasma emission when irradiating the sample with a pulse train. Each subpeak of plasma emission could be observed clearly, and the effective sub-pulses were demonstrated accurately. The details are shown in the last two paragraphs of section 2.2: the method of generating pulse trains in the frequency doubling Fabry Perot cavity.

3) From section 3.2 onwards, when the alterations from the sub-pulses train are analysed, it is reported a laser fluency parameter that is not explicated (row 189). Is that the same value as the primary Gaussian Pulse? So, the fluence before the cavity? Or is it the value of the highest energetic sub-pulse?

Thanks very much for your suggestions. First, 0.369 J/cm2 is a laser fluence measured in front of the sample target same as the primary Gaussian pulse. Second, due to the presence of the frequency-doubled Fabry-Perot cavity, there are many sub-pulses in the pulse train, 0.369 J/cm2 is a sum laser fluence of all sub-pulses. The details are shown in the first paragraph of section 3.2: Extremely high-quality LSFL induced on ITO film by pulse train.

4) Since there were used three different attenuation ratios, it would be interesting to know if the authors have calculated how many sub-pulses will affect the creation of LIPSS. As an example, I presume the attenuation ratio 5:5 would be comprised by more over-threshold sub-pulses compared to the other 2 situations.

Thanks very much for your suggestions. The details are shown in the last two paragraphs of section 2.2: the method of generating pulse trains in the frequency doubling Fabry Perot cavity.

5) It is not explained why the average depth of LSFL is higher when the Fabry-Perot cavity is used.

Thank you very much for your suggestion. In section 3.2, we described the depth and roughness of the LSFL induced by the pulse train and the primary Gaussian pulse separately, without discussing the reasons for the increase in depth. In section 3.3, we discussed the reasons for the increased depth of the LSFL: first, the hydrodynamic effect caused by the thermal effect when the primary Gaussian pulse irradiated partly erasures the LSFL. Second, the SPP excitation time is short during a primary Gaussian pulse irradiation, which is not conducive to the periodic distribution of the laser energy field, so it causes a shallow depth of the LSFL. When irradiated by the pulse train, the ablation cooling effect suppresses the thermal effect and mitigates the hydrodynamic effect on the LSFL. Second, the continuous excitation of the SPP by the pulse train allows the laser field to couple more strongly with the SPP, producing a LSFL with greater depth. The details are shown in the last paragraph of section 3.3: Discussion: The formation of extremely high-quality LSFL induced by pulse train.

6) Another important point relates section 3.4.1. Why does the laser fluence change while the effect of attenuation ratios are analysed? It goes from 0.267 J/cm2 for 5:5 and 3:7 to 0.329 J/cm2 for 1:9. Why?

Thank you for your suggestion. In our experiments, laser fluences of fabricating better LIPSS continuously increase as the attenuation ratio increases from 0.5 to 0.9. The suitable laser fluences are 0.267 J/cm2, 0.288 J/cm2 and 0.329 J/cm2 when the attenuation ratio is 0.5, 0.7 and 0.9, respectively. Demonstrating LIPSS induced by suitable laser fluence, rather than the same fluence, allows for a clearer comparison of the impact of attenuation ratio on the quality of LIPSS. The details are shown in the last paragraph of section 3.4.1: The attenuation ratio of adjacent sub-pulses.

7) Same thing happens in section 3.4.2. The fluence energy increases as the delay does. Why? And is it possible to disentangle the effect of the two parameters?

Thanks very much for your suggestion. In our experiment, laser fluences of fabricating better LIPSS continuously increase as the time interval increases from 60 to 600 ps. The reasons are as following. First, after irradiating the ITO with a presequence sub-pulse, the excited SPP decays continuously with increasing time, and the continuous excitation of SPP by the postsequence pulse decreases with the increasing of time interval. Second, the ablation cooling effect decreases as the time interval increases. Demonstrating LIPSS induced by suitable laser fluence, rather than the same fluence, allows for a clearer comparison of the impact of time interval on the quality of LIPSS.

The following minor revision comments are suggested,

1) The abstract contains a discussion about LIPSS that are more suited to be in the introduction section (row 13-16). Furthermore, the laser parameters are not labelled but just quantities are reported.

Thanks very much for your suggestion, your suggestions make sense. We have made the change marked with a red marker.

2) In the introduction, the discussion related to the HSFL is rather confusing. As an example, what is the thin-layer theory reported on row 46? Also, vertical HSFL are discussed, but what does vertical mean?

Thanks very much for your suggestion. When the sample surface was irradiated by a femtosecond laser, a thin plasma layer was excited on the surface. Thin-layer theory suggested that HSFL perpendicular to the laser polarization (SPP) was originated from surface plasmon polaritons on the lower surface of the plasma layer, or from the coupled waves of upper and lower SPP of the thin plasma layer.

Parallel means the orientation is parallel to the laser polarization, while vertical means the orientation is perpendicular to the laser polarization.

3) Since the authors in the text highlight the possibility to obtain very narrow structures (lower than 10 nm, and comparable to extreme ultraviolet lithography), it would be fitting to also report on other deep-subwavelength LIPSS obtained using different techniques to showcase the potential and the novelty of the Fabry-Perot cavity. Some examples of deep-subwavelength structures:

  1. https://doi.org/10.1021/acs.nanolett.1c01310
  2. https://doi.org/10.1364/OE.23.005357

Thank you for your suggestion, We have cited relevant literature in the article and make statement to showcase the potential and the novelty of the Fabry-Perot cavity.

[24] Mastellone, M.; Bellucci, A.; Girolami, M.; Serpente, V.; Polini, R.; Orlando, S.; Santagata, A.; Sani, E.; Hitzel, F.; Trucchi, D.M. Deep-subwavelength 2D periodic surface nanostructures on diamond by double-pulse femtosecond laser irradiation Nano Letters 2021, 21, 4477-83.

[25] Cong, J.; Yang, J.J.; Zhao, B.; Xu, X.F. Fabricating subwavelength dot-matrix surface structures of molybdenum by transient correlated actions of two-color femtosecond laser beams Optics Express 2015, 23, 5357-67.

4) In the caption of Figure 1, the BBO labelling is missing.

Thank you for your suggestion, we have included the relevant statement in the article.

5) It is not entirely clear how the laser utilization efficiency is calculated (row 105 and 106)

Thank you for your suggestion.

The laser utilization efficiency η is the ratio of the output energy Eout and the incident energy Ein as follow:

                   (1)For a transmissive Fabry-Perot cavity, if the expected energy attenuation ratio is to be 0.9 among sup-pulses, two beam splitters with transmittance of  and reflectance of  will be needed. Here the incident laser energy is Ein, energy of the first sub-pulse should be:

     (2)Energy of the second sub-pulse is:

…… (3)Energy of the nth sub-pulse is:

    (4)When n is infinitely large, the sum energy of the sub-pulses is:

…….. (5)The laser utilization efficiency η is:

……. (6) For a frequency-doubled Fabry–Perot cavity, when a laser pulse with energy Ein passes through the dichroic mirror, it will be frequency doubled by the BBO, then the laser is output only through the beam splitter, the laser utilization efficiency η is equal to the efficiency of BBO (~50%) regardless of the T:R ratio of the beam splitter.

However, a detail calculation of the laser utilization efficiency is not described due to the space limitation in the article.

6) The polarization and scanning direction of figure 3 are not entirely visible.

Thanks to your suggestion, we have made the relevant changes in the article to make the laser polarization and scanning direction more visible.

7) Have the authors also considered analyzing the width of the distribution of the spatial frequencies of 2D-FFT images? In reference n.19 the authors present, in addition to DLOA, the “dispersion” of the spatial frequencies as a characterizing tool for LIPSS’ regularity. Do the fabricated Mo samples present differences in spatial frequencies width?

Thanks to your suggestion. Widths of the distribution of the spatial frequencies of 2D-FFT images have been calculated in figures 3(b, d, f), figure 5(b) and figures 8(g, h, i) already. As described in reference: “Ten Open Questions about Laser-Induced Periodic Surface Structures”. The “dispersion” of the spatial frequencies is used as a characterizing tool for LIPSS’ regularity, its quantitative value depends on the threshold that is subjectively set. In our article, the threshold is set to be the full-width at the half of the maximum value. For example , the dispersion of the spatial frequency is ±0.192 μm-1 in figure 3(d), and ±0.040 μm-1 in figure 5(b), which indicates that more regular LSFL can be fabricated by pulse train with attenuation ratio of 0.9 and time interval of 150 ps. In figures 8(g, h, i), the dispersion is ±0.099 μm-1, ±0.071 μm-1 and ±0.040 μm-1,respectively.

Reviewer 2 Report

This work presents an interesting, though typical for parametric amplifiers, laser design for LIPSS generation with fs-laser pulse trains at limited-range variable delays. The work is very usefull, while there are sill some points for correction:

In fig.3 fluences, scan speeds and polarizaions were all varied over the three figures, which is inappropriate for their comparison (different fluences, or speeds, or polarizations).

In section 3.3 the strongly outdated views of fs-laser ablation (nanosecond or microsecond time scales, vaporization and boiling mechanisms) were presented, which should be subsantially updated regarding fs-laser removal via spallation and phase explosion - see, e.g., works [ 10.1007/s00339-013-8086-4, 10.3367/UFNe.2016.09.037974

Author Response

Comments to the Author

This work presents an interesting, though typical for parametric amplifiers, laser design for LIPSS generation with fs-laser pulse trains at limited-range variable delays. The work is very useful, while there are sill some points for correction:

Dear referee, thanks very much to you to review our manuscript, and give us a chance to revise it. Your comments are very important for us to improve this manuscript. According to your suggestions, we made revisions in the resubmitted manuscript, which is shown in blue words.

Best wishes,

All of the authors

The following major revision comments are suggested,

1) In fig.3 fluences, scan speeds and polarizations were all varied over the three figures, which is inappropriate for their comparison (different fluences, or speeds, or polarizations).

Thanks very much for your suggestions. In figure 3, the polarization and scanning directions marked in (a) are applicable to (c, e). Here we just want to show the three main morphologies of the ITO surface under the primary Gaussian pulse irradiation. In order not to cause ambiguity, we have modified the illustration of the figure accordingly.  Which is marked in blue label.

2) In section 3.3 the strongly outdated views of fs-laser ablation (nanosecond or microsecond time scales, vaporization and boiling mechanisms) were presented, which should be substantially updated regarding fs-laser removal via spallation and phase explosion see, e.g., works [ 10.1007/s00339-013-8086-4, 10.3367/UFNe.2016.09.037974].

Thank you very much for your suggestions, which are very important for our articles. The details are shown in the first paragraph of section 3.3: Discussion: The formation of extremely high-quality LSFL induced by pulse train.

New references have been added.

[30] Wu, C.P.; Zhigilei, L.V. Microscopic mechanisms of laser spallation and ablation of metal targets from large-scale molecular dynamics simulations Applied Physics A 2013, 114, 11-32.

[31] Ionin, A.A.; Kudryashov, S.I.; Samokhin, A.A. Material surface ablation produced by ultrashort laser pulses Physics-Uspekhi 2017, 60, 149-60.

Round 2

Reviewer 1 Report

The authors provided satisfactory replies to all the points raised in my review letter. However, I still think there are some minor issue they should address before pubblication.

1) While the new figure 2a makes things more clear, I still think figure 2b is incomplete. In fact, I think the authors should clarify if the first sub-pulse transmitted by the cavity has the same energy for every T:R ratio (like it is depicted in the graph), or if the incoming energy pulse is the same for every ratio, and if that is the case the graph should be probably normalized to the original gaussian pulse energy.

2) As regard point 7, I think comparing different delays between sub-pulses that ALSO have different cumulative energy is misleading for a proper comparison. The authors should explain in a more convincing way why the difference between in the energy used while also changing delay times has not any effect on the morphology.

Author Response

Comments to the Author The authors provided satisfactory replies to all the points raised in my review letter. However, I still think there are some minor issue they should address before publication. Dear referee, thanks very much to you to review our manuscript, and give us a chance to revise it. Your comments are very important for us to improve this manuscript. According to your suggestions, we made revisions in the resubmitted manuscript, which is shown in red words in this round. Best wishes, All of the authors The following minor revision comments are suggested, 1) While the new figure 2a makes things more clear, I still think figure 2b is incomplete. In fact, I think the authors should clarify if the first sub-pulse transmitted by the cavity has the same energy for every T:R ratio (like it is depicted in the graph), or if the incoming energy pulse is the same for every ratio, and if that is the case the graph should be probably normalized to the original gaussian pulse energy. Thanks very much for your suggestions. According to your suggestions, we have added statements in row 137-143 marked in red label, where “the first sub-pulse transmitted by the cavity has the same energy for every T:R ratio” has been emphasized. 2) As regard point 7, I think comparing different delays between sub-pulses that ALSO have different cumulative energy is misleading for a proper comparison. The authors should explain in a more convincing way why the difference between in the energy used while also changing delay times has not any effect on the morphology. Figure S1 LIPSS fabricating via pulse train with different delay times and laser fluences. Thanks very much for your suggestions. As shown in figure S1, figures(d–f) show similar surface morphologies corresponding to delay times and laser fluences of 300 ps / 0.33 J/cm2, 400 ps / 0.33 J/cm2, 600 ps / 0.33 J/cm2. When the laser fluence is kept at 0.33 J/cm2, the quality of LSFL are much worse than those of figures (g-i). It is a common phenomenon that under different attenuation ratio or delay time, the proper laser fluence of fabricating regular LIPSS is different. Here are the reasons: First, under different attenuation ratio, pulse train with attenuation ratio of 0.9 has the lowest sub-pulse energy if total laser fluences stay the same compared to 0.5 and 0.7, the fabricating processing of LIPSS in many cases is ablative, that is, in order to fabricate LIPSS, the first sub-pulse energy need to reach the ablation threshold of ITO film, which means the proper laser fluence with attenuation ratio of 0.9 to fabricate LIPSS is higher than those of 0.5 and 0.7. Second, under different delay time, the laser ablation threshold is different due to the ultrafast process(i.e. electron dynamics, phase explosion, heat transfer. etc) of ultrafast laser ablation. In the new reference we added [10.1016/j.apsusc.2020.147940], the effect of second pulse to the first pulse significantly differs as the time delay increase from 3 ps to 30 ps, which causes the increasement of ablation threshold. In our article, we aim to fabricate LIPSS of high quality, so in the comparison of LIPSS with different time delay, we choose LIPSS with relative better quality rather than LIPSS with same fluences. We have added statement and new reference in row 343-347. [39] Kudryashov, S.I.; Samokhvalov, A.A.; Golubev, Y.D.; Ivanov, D.S.; Garcia, M.E.; Veiko, V.P.; Rethfeld, B.; Mikhailovskii, V.Y. Dynamic all-optical control in ultrashort double-pulse laser ablation Appl. Surf. Sci. 2021, 537, 147940. doi: 10.1016/j.apsusc.2020.147940 Please write down "Please see the attachment

Reviewer 2 Report

Since formtaion of LIPSS is ablative in many cases (not only translative) and "laser fluences of fabricating better LIPSS continuously increase 328 as the attenuation ratio increases from 0.5 to 0.9" (page 10, lines 328-329), the variable interpulse delays (60-1000 ps) could extend till plme screening times (>100 ps, see, e.g., [10.1016/j.apsusc.2020.147940]) and this effect should be considered in the study.

Another minor issue is to keep just the meaningfull digits in the experimental data values across the manuscript - see, e.g., fluences  0.267 J/cm2 = 0.27 J/cm2? DSOA 15.58° = 15.6°? Fourier transform peak is at 2.300 ± 0.040 μm-1, corresponding to a period of 435 ± 8 nm? etc. Please, evaluate the data acquisition accuracy and use it to present your data. 

Author Response

Dear referee, thanks very much to you to review our manuscript, and give us a chance to revise it. Your comments are very important for us to improve this manuscript. According to your suggestions, we made revisions in the resubmitted manuscript, which is shown in blue words in this round. Best wishes, All of the authors The following minor revision comments are suggested, 1) Since formtaion of LIPSS is ablative in many cases (not only translative) and "laser fluences of fabricating better LIPSS continuously increase as the attenuation ratio increases from 0.5 to 0.9" (page 10, lines 328-329), the variable interpulse delays (60-1000 ps) could extend till plume screening times (>100 ps, see, e.g., [10.1016/j.apsusc.2020.147940]) and this effect should be considered in the study. Thanks very much for your suggestions. Your in-depth suggestion is really important not only to the article but to our further study. The proper laser fluences of fabricating better LIPSS vary with not only attenuation but time delays due to the ablative fabricating processing of LIPSS in many cases. It is related to the ablation threshold as you mentioned in the reference which was written: “If the second pulse delay is longer, however, the laser-induced stresses due to the first pulse have enough time for relaxation and the second pulse therefore will be absorbed by acoustically relaxed material, which is already in the process of hydrodynamic motion…….. The resulting removed volume of the material is less than it is for the case of 3 ps second pulse delay and the ablation threshold for the pulse delay of 30 ps is therefore increased.” Indeed, the effect of the time delay and the attenuation ratio to the ultrashort process of ultrafast laser ablation is more complicated and need to be further studied. We suggest preparing ultrafast luminescence experiments to explore this critical problem. When irradiating the sample with a pulse train, a streak camera can be used to detect plasma emission. Each sub-peak of plasma emission might be clearly observed. In the article, we have added description to explain the increase of laser fluences when the attenuation ratio increases from 0.5 to 0.9 in row 332-334 marked in blue label. For the variation of laser fluence with the time delay, statement has been added to draw the readers’ attention to relevant reference. New reference has been added in row 346-348 which marked in blue label. [39] Kudryashov, S.I.; Samokhvalov, A.A.; Golubev, Y.D.; Ivanov, D.S.; Garcia, M.E.; Veiko, V.P.; Rethfeld, B.; Mikhailovskii, V.Y. Dynamic all-optical control in ultrashort double-pulse laser ablation Appl. Surf. Sci. 2021, 537, 147940. doi: 10.1016/j.apsusc.2020.147940 2) Another minor issue is to keep just the meaningfull digits in the experimental data values across the manuscript - see, e.g., fluences 0.267 J/cm2 = 0.27 J/cm2? DSOA 15.58° = 15.6°? Fourier transform peak is at 2.300 ± 0.040 μm-1, corresponding to a period of 435 ± 8 nm? etc. Please, evaluate the data acquisition accuracy and use it to present your data. Thank you very much for your suggestions. We have checked the data and made relevant changes according to your suggestion.
